# Attack of the dark clones the genetics of reproductive and color traits of South African honey bees (*Apis mellifera* spp.)

Laura Patterson Rosa[1‡]*, Amin Eimanifar[2‡], Abigail G. Kimes[3], Samantha A. Brooks[3,4‡], James D. Ellis[1‡]

1 Honey Bee Research and Extension Laboratory, Entomology and Nematology Department, University of Florida, Gainesville, Florida, United States of America, 2 Independent Senior Research Scientist, Industrial District, Easton, Maryland, United States of America, 3 Department of Animal Sciences, University of Florida, Gainesville, Florida, United States of America, 4 UF Genetics Institute, University of Florida, Gainesville, Florida, United States of America

‡ LPR and AE are first co-authors and contributed equally to the work and SAB and JDE are co-senior authors and contributed equally to the work.
* lpattersonr@ufl.edu

**Data Availability Statement:** Genotype data utilized in this publication are available in Eimanifar, Brooks, Bustamante, & Ellis, 2018 (https://doi.org/10.1186/s12864-018-4998-x). Phenotype data are

## Abstract

The traits of two subspecies of western honey bees, *Apis mellifera scutellata* and *A.m. capensis*, endemic to the Republic of South Africa (RSA), are of biological and commercial relevance. Nevertheless, the genetic basis of important phenotypes found in these subspecies remains poorly understood. We performed a genome wide association study on three traits of biological relevance in 234 *A.m. capensis*, 73 *A.m. scutellata* and 158 hybrid individuals. Thirteen markers were significantly associated to at least one trait ($P \leq 4.28 \times 10^{-6}$): one for ovariole number, four for scutellar plate and eight for tergite color. We discovered two possible causative variants associated to the respective phenotypes: a deletion in *GB46429* or *Ebony* (NC_007070.3:g.14101325G>del) (R69Efs*85) and a nonsense on *GB54634* (NC_007076.3:g.4492792A>G;p.Tyr128*) causing a premature stop, substantially shortening the predicted protein. The mutant genotypes are significantly associated to phenotypes in *A.m. capensis*. Loss-of-function of *Ebony* can cause accumulation of circulating dopamine, and increased dopamine levels correlate to ovary development in queenless workers and pheromone production. Allelic association ($P = 1.824 \times 10^{-5}$) of NC_007076.3:g.4492792A>G;p.Tyr128* to ovariole number warrants further investigation into function and expression of the *GB54634* gene. Our results highlight genetic components of relevant production/conservation behavioral phenotypes in honey bees.

## Introduction

Modern western honey bees (*Apis mellifera*) show substantial genetic and phenotypic variation across their extensive geographic range [1]. They occur naturally in Europe, the Middle East, western Asia, and Africa, where the species is composed of between 25–35 subspecies [2–4]. This bee has been spread outside its native range to the Americas, Australia, New Zealand, and

available in Bustamante, Baiser & Ellis, 2020
(https://doi.org/10.1007/s13592-019-00651-6).
Individual genotype data are available through
Dryad Digital Repository (https://doi.org/10.5061/
dryad.98jh446).

**Funding:** This work was supported through various
cooperative agreements provided by the United
States Department of Agriculture, Animal and Plant
Health Inspection Service and by the Florida
Department of Agriculture and Consumer Services
through the guidance of the Honey Bee Technical
Council. The funders had no role in study design,
data collection and analysis, decision to publish, or
preparation of the manuscript.

**Competing interests:** The authors have declared
that no competing interests exist.

other locations globally, largely due its ability to produce honey and its use as the principal pollinator of a variety of agricultural crops.

Two subspecies of western honey bees, *A.m. scutellata* and *A.m. capensis*, are among those endemic to the Republic of South Africa (RSA) [5]. *Apis mellifera scutellata* is a light-colored phenotype and is adapted for survival in hot and arid climates in central and southern Africa [6]. It also displays behavioral traits that many beekeepers outside its native range consider undesirable. These include excessive swarming (colony-level reproduction), absconding (complete nest abandonment), usurpation (swarm takeover of another colony) and heightened defensiveness [6–8]. This honey bee subspecies was introduced into Brazil in the 1950's in an effort to improve the Brazilian beekeeping industry [9]. It hybridized with local stocks of European-derived honey bees, becoming known as "Africanized" or "killer" bees. They are now considered invasive throughout South America, Central America and southern regions of North America [7, 10].

*Apis mellifera capensis* is a darker colored honey bee subspecies found in the Fynbos region of RSA, where the climate is Mediterranean with rainy winters. In contrast to *A.m. scutellata*, this bee can act as a social parasite, given its workers can reproduce via thelytoky [6], a type of parthenogenesis in which female offspring can result from unfertilized eggs. This trait allows some worker bees to develop into pseudoqueens with semi-developed spermathecae, that remain unused, and a larger-than-normal number of ovarioles [11–15]. These worker bees, then, can fly into neighboring hives and replace the queens contained within, becoming the reproductive in the nest [16]. Interestingly, colonies headed by *A.m. capensis* workers are doomed, as laying workers cannot maintain the egg output of that of a normal queen. The colonies eventually dwindle and die, resulting in the '*capensis* calamity' that has plagued the South African beekeeping industry in the past [17].

Despite the perceived drawbacks associated with these bees outside their native range, beekeepers in RSA keep both subspecies for management purposes. Nevertheless, the potential movement of both bee subspecies beyond where they currently occur remains a concern of beekeepers and regulatory officials in many areas globally. These concerns have led to the search for better methods to identify both bee subspecies and their hybrids quickly and reliably. Recently developed techniques based on the reduction of genome complexity, such as Genotyping by Sequencing (GBS), have the potential to provide a large number of SNPs in understudied genomes, enabling genetic diagnostics for monitoring these two subspecies [18]. Despite genomic studies on various honey bee subspecies, the genetic basis of important phenotypes found in *A.m. scutellata* and *A. m. capensis* remain poorly understood, though progress has been made with the thelytoky trait [19–25]. We have the opportunity to fill this gap given recent work [26] that used traditional morphometric techniques to identify populations of both bees from samples collected in RSA.

In the present study, we performed a genome wide association study (GWAS) on three traits (number of ovarioles, tergite and scutellar plate color) measured in 464 *A.m. capensis*, *A. m. scutellata* and hybrid individuals (S1 Table) [26]. These same bees had been examined previously using GBS [18]. *Apis mellifera capensis* is known to be darker and have a greater number of ovarioles per ovary than does *A.m. scutellata*. Accordingly, the GWAS allowed us to determine what chromosomal regions are most associated with these phenotypic traits. The detected associations provide improved understanding of the genetic basis of phenotypic and behavioral differentiation between *A.m. capensis* and *A.m. scutellata* from RSA.

## Results

### GWAS associates traits mainly to two chromosomes

The GWAS resulted in significant associations to markers on chromosomes LG1, LG2, LG7, LG9 and LG10. Thirteen markers were significantly associated to at least one trait ($P \le 4.28$ x

$10^{-6}$): one for ovariole number, four for scutellar plate and eight for tergite color. A total of 10 genes are annotated in candidate regions determined by markers within $r^2 \geq 0.2$ to the most significant marker, and adjacent genes (Fig 1 and Table 1).

## A frameshift and a nonsense mutation are associated to color and ovariole number

Functional inspection of annotated genes within each candidate region indicated two genes with coding variants. The likely candidate gene for tergite and scutellar plate color is *GB46429*, *mycosubtilin synthase subunit C*, also known as *Ebony*, a non-ribosomal peptide synthetase, which also has sequence similarities to microbial enzymes [27]. This gene shares 46.99% (EnsemblMetazoa release 103, LOC409109) [28] of its sequence with the *Drosophila melanogaster Ebony* gene. A deletion identified by the GBS pipeline in *GB46429* (NC_007070.3: g.14101325G>del) (R69Efs*85) leads to an early stop codon and truncates the normal amino acid sequence from the predicted 860aa to only 85 amino acids.

A single variant was found for ovariole number within the coding region of *GB54634*. The nonsense SNP (NC_007076.3:g.4492792A>G;p.Tyr128*) causes a premature stop, shortening the protein by two of the six predicted exons (45% of the protein sequence) (Fig 2).

The distribution of causative variants demonstrates that the mutant form is significantly associated to phenotypes in *A.m. capensis*, while the wildtype locus is associated to *A.m. scutellata* phenotypes (Fig 3). No coding variants were discovered in our GBS dataset for the other annotated genes within each candidate region; yet these could hold biological effects of interest for the honey bee.

## Discussion

Color variation in honey bees may have diverse biological implications [30]. For example, Gloger's rule states that coloration changes according to environmental effects, and species tend to

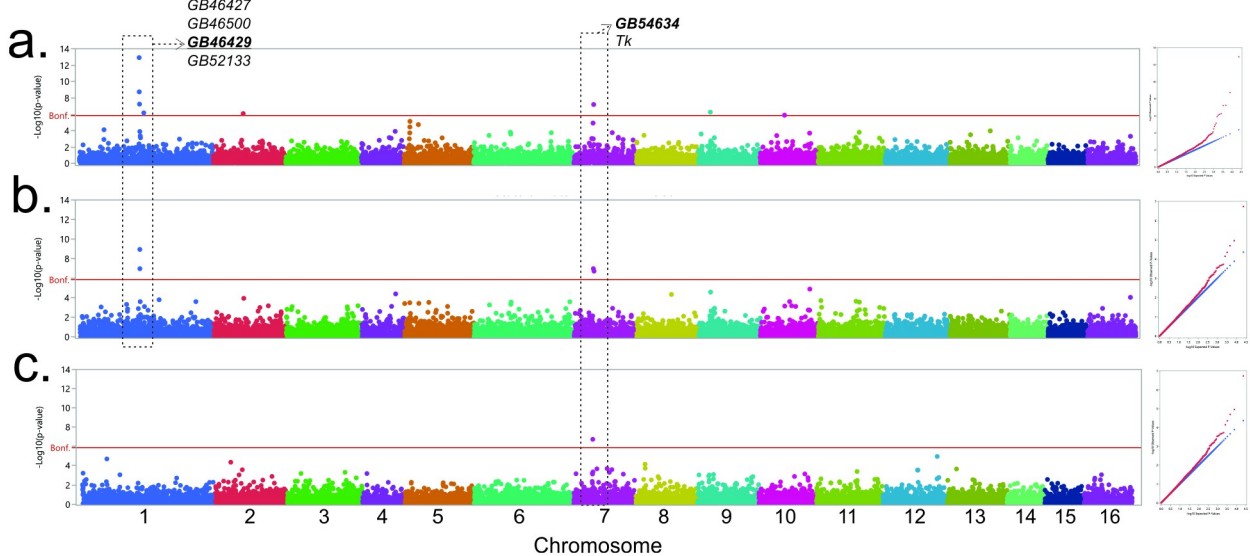

**Fig 1. Manhattan and QQ plots of the respective genome wide association study for a. tergite color ranked-transformed; b. scutellar plate color ranked-transformed; and c. ovariole number rank-transformed.** Respective annotated genes within the shared regions in chromosomes LG1 and LG7, as well as genes possessing non-synonymous variants (in bold), are also shown. The red line represents the Bonferroni corrected threshold value of $P \leq 4.28 \times 10^{-6}$, and markers above this line are significantly correlated to the respective trait.

**Table 1. Genome wide association study traits, significant markers, respective chromosome (Chr) location, number of base pairs, statistical information, and within region/nearby annotated genes.**

| Trait | Marker ID | Chr | Base Pair | P-value | Genes |
|---|---|---|---|---|---|
| **Ovariole Number** | S1_108729877* | 7 | 4497718 | 1.8241 x 10−7 | *GB54634* |
| | | | | | *Tk* |
| **Scutellar Plate** | S1_14074192** | 1 | 14074192 | 1.12149 x 10−9 | *GB46427 GB46500 GB46429* |
| | S1_14077754** | 1 | 14077754 | 1.03746 x 10−7 | |
| | S1_108729877* | 7 | 4497718 | 1.03692 x 10−7 | *GB54634* |
| | | | | | *Tk* |
| | S1_108786716*** | 7 | 4554557 | 1.83841 x 10−7 | − |
| **Tergite Color** | S1_14074192** | 1 | 14074192 | 1.19236 x 10−13 | *GB46427 GB46500 GB46429* |
| | S1_14077754** | 1 | 14077754 | 1.79334 x 10−9 | |
| | S1_14080360** | 1 | 14080360 | 5.6368 x 10−8 | |
| | S1_15042195 | 1 | 15042195 | 6.60343 x 10−7 | *GB52133* |
| | S1_34570644 | 2 | 4677136 | 8.12408 x 10−7 | − |
| | S1_108786716*** | 7 | 4554557 | 6.28334 x 10−8 | − |
| | S1_132740677 | 9 | 1742429 | 5.37694 x 10−7 | *GB43750 GB43751 GB43755 Tpx-4* |
| | S1_147958987 | 10 | 5840186 | 1.25779 x 10−6 | − |

*An asterisk (*) represents the marker sharing the same candidate region.

⁻A minus (−) means no annotated genes occurred in the candidate region.

be darker in hot and humid environments [31]. Yet, this rule might not apply to the present case, as the *A.m. scutellata* individuals were collected from, on average, warm semi-arid zones, while the *A.m. capensis* or hybrid samples came from cooler, Mediterranean or cool subtropical zones; yet, *A.m. scutellata* had significantly lighter phenotypes both in tergite and scutellar plate color [26]. Additionally, previous thelytoky genome mapping efforts pointed to a locus near *GB46429* [24]. However further inspection into expression and gene function demonstrated that this gene has no apparent effect on the mode of parthenogenesis in the honey bee, but segregates according to subspecies and color [32].

In *Drosophila melanogaster*, the orthologous gene to *GB46429* is *Ebony* (named after the mutant phenotype): darker *Drosophila* flies have lower expression, while lighter individuals have normal to high expression of *Ebony* [33]. Variants in *Ebony* also contribute to diverse phenotypic variations including behavioral, neurologic, locomotor, and visual ability [34, 35]. Some *Drosophila Ebony* mutants' electroretinograms lacked the on- and off-transients of light response [36, 37]. Most importantly, *Ebony* participates in dopaminergic neuron function, metabolizing dopamine into *N*-β-alanyl dopamine (NBAD) [38]. We discovered a single nonsense variant in *GB46429 (Ebony)* significantly associated to color phenotypes in both honey bee subspecies and hybrids of the two. Furthermore, this variant severely impacts the predicted protein structure and may lead to loss-of-function of this protein. Consistent with our findings, loss-of-function *Ebony* mutants in *Drosophila* accumulate circulating dopamine, which is then directed to other pathways [39]. In the honey bee, increased dopamine levels correlate to ovary development in queenless workers, as the queen mandibular pheromone (QMP) regulates dopamine pathways in the worker bees [40, 41]. In *A.m. capensis*, pheromonal dominance allows for parasitic behavior, even in the presence of an *A.m. scutellata* queen [42]. We postulate that the mutant *GB46429* causes a darker pigmentation phenotype and may play a role in dopaminergic pathways and parasitic behavior in *A.m. capensis*. This gene's contribution to behavioral and reproductive traits in honey bees is worthy of further investigation.

## a. *GB46429/Ebony* ("Color") protein

Wildtype Mutant

## b. *GB54634* ("Ovariole Number") protein

Wildtype Mutant

**Fig 2. Predicted protein structure (Phyre2) for both wild type and discovered variants of genes.** a. *GB46429 (Ebony)*, correlated to both scutellar plate and tergite color phenotypes., and b. *GB54634*, correlated to ovariole number.

Previous work evaluating quantitative trait loci (QTLs) impacting the number of ovarioles in honey bees resulted in a significant QTL on LG11 [43]. Although our GWAs did not associate any markers on LG11 to ovariole number, this difference in findings could be due to population genetic differences as the LG11 QTL resulted from Africanized Honey Bees (AHB) collected in Arizona, USA, compared to European Honey Bee samples collected from US commercial colonies.

Unfortunately, there is little information of the function and expression of the *GB54634* gene in honey bees even though we found the significant ($P = 1.824 \times 10^{-5}$) allelic association of the NC_007076.3:g.4492792A>G;p.Tyr128* variant to ovariole number (Fig 3). The *GB54634* gene was tagged by genomic sweeps associated to social parasitic behavior [44] and *A.m. capensis* versus *A.m. scutellata* differentiation [18], although candidate genes for thelytoky phenotype recently reported do not implicate *GB54634* in this specific phenotype [24, 25]. Additionally, this uncharacterized protein (*LOC725260 isoform X1*) does not seem to differ in expression and splicing in the presence or absence of queen pheromones [45]. Yet, this expression analysis was conducted in an uncharacterized *A. mellifera* subspecies; thus, findings could be different for *A. m. capensis* and *A. m. scutellata*. Given the correlation reported here, further investigation into association of this variant to social parasitic traits such as the number of ovarioles, as well as possible pleiotropic effects, warrants additional exploration and biological characterization of *GB54634*.

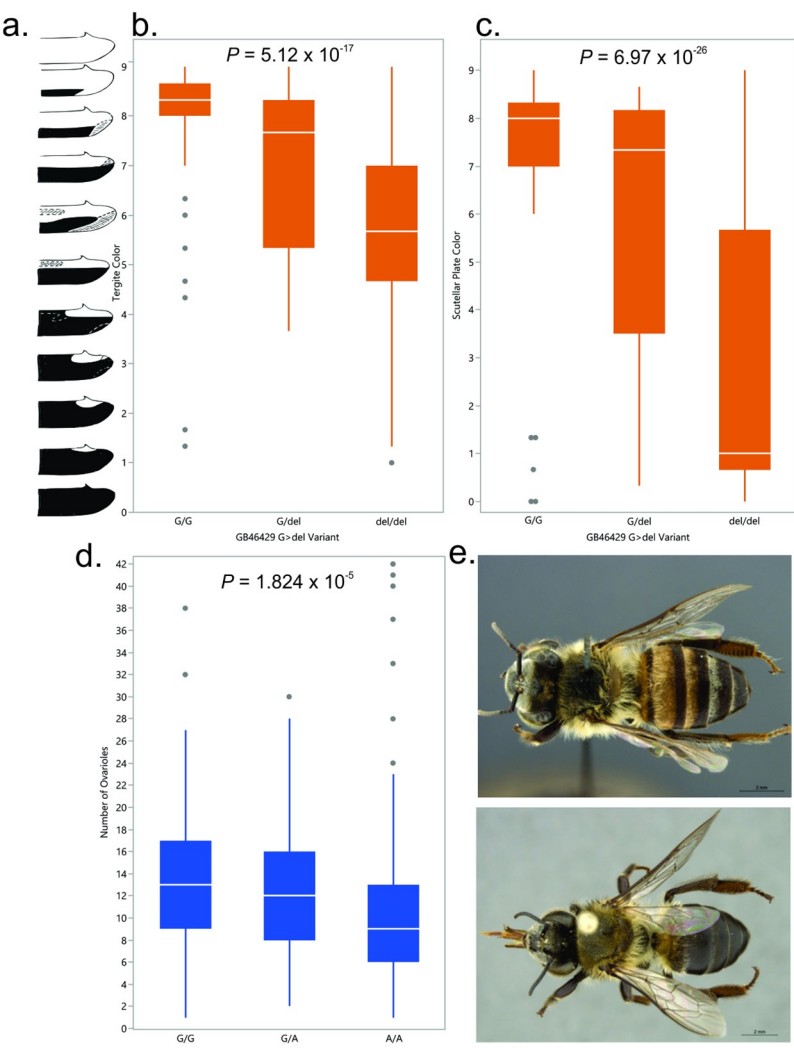

**Fig 3. Allele distribution for variants discovered in *GB54634* and *GB46429 (Ebony)*, as well as respective color phenotypes.** a. Ruttner [2] ranking for tergite color, also applied to scutellar plate phenotyping [19]; b. Allelic distribution of the NC_007070.3:g.14101325G>del;p.R69Efs*85 variant for tergite color and c. scutellar plate color; d. Allelic distribution of the NC_007076.3:g.4492792A>G;p.Tyr128* variant for ovariole number; and e. Individuals from the *Apis mellifera scutellata* (above) and *A.m. capensis* (below) representing the variation in color [29].

Although we did not discover coding sequence variants for the other genes within candidate regions, biological functions related to *A. m. capensis* phenotypes may be of interest for future analysis. For instance, candidate regions for the color traits *GB43750* (*prefoldin subunit 5*) are located within a haplotype associated to high altitude adaptation in *A. m. scutellata* [46].

*TK* (prepro-AmTRP or tachykinin) was also found in the candidate region associated to ovariole number in the GWAS. Previously implicated in female-related behavior, the expression levels of prepro-AmTRP are present only in the brain of female bees (queens and workers) and show lower expression levels according to labor division (lower in younger/nurse bees, higher in queens and foragers) [47]. The tachykinin neuropeptide also controls metabolic and desiccation responses in *Drosophila* [48, 49] and is related to aggression in other insects, such as the Leaf-Cutting Ant *Acromyrmex echinatior* [50]. Other AmTRP neuropeptides are implicated in the defensive behavior of Africanized honey bees [51].

Several genomic regions are likely involved in ovariole number, a social parasitism-related phenotype of *A.m. capensis* colonies [44]. The *GB46427* gene (*LOC409096*) within the ovariole number LG1 candidate region is implicated in parasitism behavior and was deemed the thelytoky gene [24], also demonstrating Log2-fold differential expression of 3.24 to 4.68 between thelytokous *A.m. capensis* and arrhenotokous *A.m. scutellata* [24, 44]. A non-synonymous variant (p.Thr400Ile) likely responsible for this differential expression was suggested as the sole change responsible for thelytoky in worker bees [24]. Our GBS dataset did not possess any variants within the coding region of this gene; thus, we could not evaluate the phenotypic repercussions. Furthermore, *GB46500* (*LOC724495* or *Ethr*) is also in linkage with *GB46427* [24]. In *Drosophila*, lower levels of the hormones transcribed by *Ethr* halt oogenesis and ovulation during nutritional or heat stress [52]. Therefore, its effects on honey bee social parasitism might be of biological relevance, though we could not find coding variants for this gene.

## Conclusions

We associated genomic regions with important biological phenotypes as tergite color, scutellar plate color, and ovariole number within *A.m. capensis* and *A.m. scutellata* populations from RSA. Among the 28 candidate genes identified, *Ebony*, within the tergite color candidate region on chrLG1, possessed a variant predicted to alter protein structure significantly. Furthermore, non-functional variants of *Ebony* impacting pigmentation are well-documented in other insect species. Although the candidate variant correlated to ovariole number is in an uncharacterized gene, further investigations into its function are warranted given its biological implications. Our results help pave the way for the development of marker-assisted selection and diagnostic genetic differentiation in the honey bee and highlight potentially production/ conservation relevant pleiotropic behavioral phenotypes.

## Material and methods

### Honey bee samples

The samples included 464 adult worker honey bees collected from RSA in 2013 and 2014. The samples were collected from managed colonies of *A. mellifera* with permission granted by the owner beekeepers (see Acknowledgements). Location data for the samples, including GIS coordinates, can be found in S1 File. Combined morphometrics, SNP, microsatellite and mitochondrial DNA data were used to determine that 73 bees were *A.m. scutellata*, 234 were *A.m. capensis* and 158 were hybrids of the two subspecies [18, 26, 53, 54]. Phenotyping methods as described in [19] determined morphometric phenotypes that significantly differed between the two subspecies of honey bees. We utilized the following traits in a GWA: number of ovarioles, pigmentation of abdominal tergite (A3) and pigmentation of the scutellar plate (Fig 4 and S1 File). The distribution for these quantitative phenotypes within the 464 samples was not normal; thus, we normalized the data prior to the GWAs using a Rank normalization on JMP®, Version 15 (SAS Institute Inc., Cary, NC, 1989–2019).

### Genotyping and SNP QC

DNA extraction, library construction, sequencing and quality control criteria were conducted by The Genomic Diversity Facility at Cornell University. The GBS methods were previously described [18], and resulted in an average of 70,475 SNPs per individual sample. We filtered GBS SNPs (coded as major/minor allele) using VCFtools version 0.1.15 [55] and the following criteria: (1) no more than two alleles, neither of which was a gap allele, (2) a minor allele frequency (MAF) of at least 5%, (3) no more than 92% missing data, (4) mapped to one of the 16

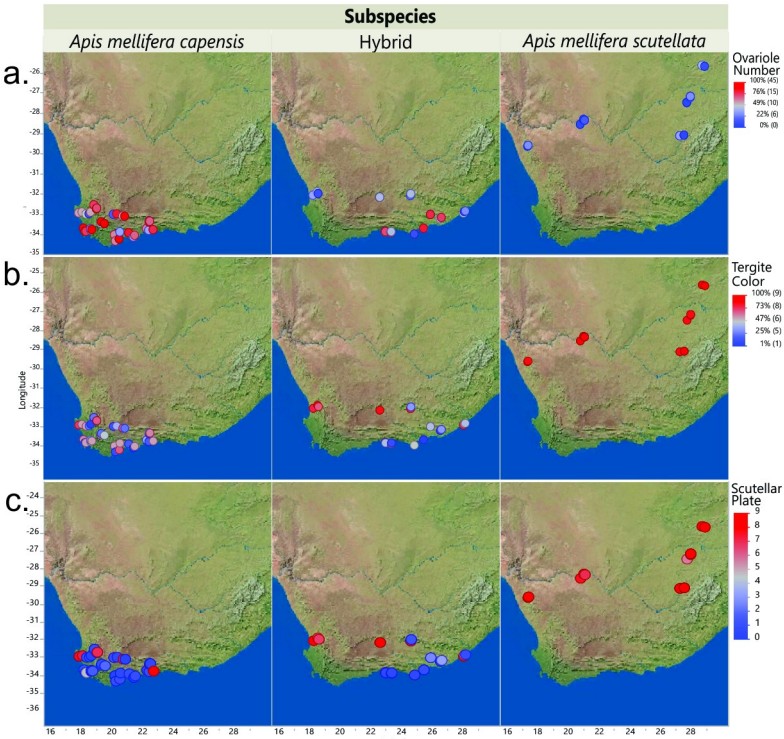

**Fig 4. Distribution of morphometric phenotypic traits per subspecies, representing a. Ovariole Number quantile, b. Tergite Color quantile and c. Scutellar Plate color ranking.** Red represents increased number of ovarioles (a) or lighter phenotypes (b and c), while blue represents lower number of ovarioles (a) and darker phenotypes (b and c). The visual distribution seems to correlate to the subspecies or hybrid geographical distribution.

assembled *A. mellifera* chromosomes in the Amel4.5 assembly [56] and (5) with an index of panmixia ($F_{IT}$) of at least −0.2. After quality control, 20,006 SNPs were left. We then imputed missing genotypes for the 20,006 loci using Beagle 4.1 [57] with a window and overlap of 500 and 50 sites, respectively. After imputation, the SNPs were again filtered for MAF of at least 5%, resulting in a total of 11,656 SNPs retained per individual. The resulting VCF file was converted to PLINK format/binary ped format with the—recode—make-bed command in PLINK version 1.90b3.39 [58].

## Genome wide association study

We performed a GWAS using a mixed linear model (MLM) analysis with the interrogated SNPs falling on the same chromosome as the given candidate SNP excluded from the genetic relationship matrix calculation (—mlma-loco) in GCTA ver. 1.25.2 [59]. A genetic relationship matrix (GRM) was included in the MLM analysis to compensate for population structure within the sample. We utilized a Bonferroni corrected threshold of $P \leq 1.429 \times 10^{-6}$ as the significance cutoff based on 11,656 SNPs tested and the three traits analyzed ($\alpha$ = 0.05). We visualized GWA results in JMP$^{\circledR}$, Version 15 (SAS Institute Inc., Cary, NC, 1989–2019).

## Identification of candidate genes and functional variants

Markers above Bonferroni correction were inspected for supporting linkage ($r^2$) in PLINK (—chr [ChromosomeNumber]—r2—ld-snp [MarkerID]—ld-window-r2 0.00—ld-window 100000). Loci with a $r^2 \geq 0.2$ to the lowest p-value SNP defined the boundaries of candidate

regions considered for further analysis [60, 61]. We also evaluated genes adjacent to each candidate region, determined using the NCBI/GenBank annotation GCF_000002195.4 /GCA_000002195.1 (Amel 4.5) [56]. Gene function was also reported based on its homology to functionally characterized genes from the *A. mellifera* genome (Amel 4.5) using the EnsemblMetazoa database (release 103) [28] and a comprehensive scientific literature search on other Hymenoptera order members [62, 63].

Visual inspection of genomic regions for polymorphisms within coding regions was performed on the unfiltered, not imputed, GBS generated. vcf file, aligned, and uploaded to NCBI *Apis mellifera* 4.5 (accession number GCF_000002195.4), coded as major/minor allele. For candidate mutations, we evaluated protein impact using Phyre2, modeling both the wild type and the sequence containing mutation(s) [64]. Allelic association of causative polymorphisms to traits was performed on JMP®, Version 15 (SAS Institute Inc., Cary, NC, 1989–2019) using ANOVA, with the significance threshold set to $P \leq 0.00833$ based on multiple tests per allele (0.05/6).

## Supporting information

**S1 File. Phenotypic information, geographical coordinates and candidate variant genotypes for samples used in this study.** Ovary number value, scutellar plate and tergite color scores and respective rank transformations, as well as respective combined probable *Apis mellifera* subspecies ID and candidate variant genotypes per sample.
(XLSX)

**S2 File. Candidate variant distribution of alleles per subspecies.**
(PDF)

**S1 Table. Sample *Apis mellifera* subspecies assignment per source information.** Hybrid = A cross between *A.m. scutellata* and *A.m. capensis*. NA = bee samples from that location were not included in the respective analysis. The "Combined Probable ID" is inferred from the most common identification (ID) made across the four referenced studies and it parallels the identifications assigned using SNPs.
(PDF)

## Acknowledgments

We thank current and former members of the University of Florida Honey Bee Research and extension Laboratory who collected honey bee samples across the Republic of South Africa: Tomas Bustamante, Mark Dykes, Ashley Mortensen, and Daniel Schmehl. We also thank Mathias Ellis for assistance with sample collection. We graciously acknowledge Mike Allsopp (ARC-Plant Protection Research Institute, RSA), Christian Pirk (University of Pretoria, RSA), and Garth Cambray for the assistance they provided in coordinating field sample collections and/or providing samples. We also thank the RSA beekeepers who allowed us to sample their colonies. We thank Dr. Ann Staiger for all the scientific input that improved the outcome of this research.

## Author Contributions

**Conceptualization:** Laura Patterson Rosa, Amin Eimanifar, Samantha A. Brooks, James D. Ellis.

**Data curation:** Laura Patterson Rosa, Abigail G. Kimes.

**Formal analysis:** Laura Patterson Rosa, Abigail G. Kimes, James D. Ellis.

**Funding acquisition:** James D. Ellis.

**Investigation:** Laura Patterson Rosa, Amin Eimanifar, Abigail G. Kimes, Samantha A. Brooks.

**Methodology:** Laura Patterson Rosa, Samantha A. Brooks.

**Project administration:** Samantha A. Brooks, James D. Ellis.

**Resources:** James D. Ellis.

**Software:** Samantha A. Brooks.

**Supervision:** Samantha A. Brooks, James D. Ellis.

**Validation:** Laura Patterson Rosa, Abigail G. Kimes.

**Visualization:** Laura Patterson Rosa.

**Writing – original draft:** Laura Patterson Rosa, Amin Eimanifar, Abigail G. Kimes.

**Writing – review & editing:** Laura Patterson Rosa, Amin Eimanifar, Abigail G. Kimes, Samantha A. Brooks, James D. Ellis.

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
