## [Decision Letter · Decision Letter 0]

8 Oct 2021

PONE-D-21-27869Attack of the Dark Clones The genetics of reproductive and color traits of South African honey bees (Apis mellifera spp.)PLOS ONE

Dear Dr. Patterson Rosa,

Thank you for submitting your manuscript to PLOS ONE. The reviewers and I found the topic and results of this study interesting but there are several concerns that need to be addressed. Most importantly, the major concerns raised by the second reviewer about samples, data availability, and the impact of imputation are certainly critical to address, in addition to the other concerns of both reviewers. Thus, while I decided to recommend "minor revisions" because the changes in the manuscript itself would not be extensive, this should certainly not be mistaken as a decision that publication can be guaranteed at this point.

We look forward to receiving your revised manuscript.

Kind regards,

Olav Rueppell

Academic Editor

PLOS ONE

2. In your Methods section, please provide additional details regarding the source of the bees used in your study. If they was collected please include geographic coordinates of your field collection site if available and provide the permits you obtained for the work. Please ensure you have included the full name of the authority that approved the field site access and, if no permits were required, a brief statement explaining why. If this was purchased then please include the names of the purchasing sources (e.g., stores, markets, suppliers), if available, as well as any further details about the purchased items (e.g., lot number, source origin, description of appearance) to ensure reproducibility of the analyses. For more information regarding PLOS' policy on materials sharing and reporting, see https://journals.plos.org/plosone/s/materials-and-software-sharing#loc-sharing-materials.

“This work was supported through various cooperative agreements provided by the United States Department of Agriculture, Animal and Plant Health Inspection Service and by the Florida Department of Agriculture and Consumer Services through the guidance of the Honey Bee Technical Council. The funders had no role in study design, data collection and analysis, decision to publish, or preparation of the manuscript.”

Reviewers' comments:

Reviewer's Responses to Questions

**Comments to the Author**

1. Is the manuscript technically sound, and do the data support the conclusions?

Reviewer #1: Yes

Reviewer #2: Yes

2. Has the statistical analysis been performed appropriately and rigorously? 

Reviewer #1: Yes

Reviewer #2: No

3. Have the authors made all data underlying the findings in their manuscript fully available?

Reviewer #1: Yes

Reviewer #2: No

4. Is the manuscript presented in an intelligible fashion and written in standard English?

Reviewer #1: Yes

Reviewer #2: Yes

5. Review Comments to the Author

Reviewer #1: This is an interesting paper on a genome wide association study in Apis mellifera capensis, Apis mellifera scutellata and hybrid individuals looking for associations to the number of ovarioles and two color-related traits: tergite and scutellar plate color. The authors discovered two possible genetic variations associated to these traits, a deletion in the ebony gene and a premature stop on gene GB54634.The experiments seem to be performed accurately and are well presented. The subject is put into the context of similar studies and is discussed appropriately. In my opinion, the manuscript would make a useful addition to the literature. Only a few minor suggestions to further improve the manuscript came into my mind.

Specific comments:

Minor issues:

line 102, “Bustamente et al. (2020)”: Enter the reference number as in the references list! (21)? See also line 249.

line 108, “Eimanifar et al. (2018a)”: Reference number! (18)?

line 137 or elsewhere: When the Ebony protein is introduced, it should be mentioned that Ebony is a non-ribosomal peptide synthetase, which also has sequence similarities to microbial enzymes (Hovemann et al., 1998).

lines 144/145, line 147, “Figure 2”: Check Figure numbers! Protein structures are given in Figure 3.

line 153, line 157, “Figure 3”: Check Figure numbers! Description matches Figure 2.

lines 158/159, “Ruttner (1988)” and “Bustamente et al., (2020)”: Enter reference numbers as in the references list! (2) and (21), respectively?

lines 177-178, Variants in Ebony also contribute to diverse phenotypic variations including … visual ability: In fact, it was shown some time ago that the electroretinogram (ERG) of certain Drosophila ebony mutants lacked the on- and off-transients of the light response (Hotta and Benzer, 1969; Heisenberg, 1971).

lines 178-180, “Ebony participates in dopaminergic neuron function, metabolizing dopamine into N-β-alanyl dopamine (NBAD)”: That is correct, but not the whole truth. Drosophila ebony mutants also have reduced histamine content in the head and are unable to convert histamine into its β-alanine conjugate, carcinine (Borycz et al., 2002). In vitro assays with heterologously expressed and purified Drosophila Ebony protein have shown later that the activation and binding of β-alanine occurs in a peptide-synthetase like manner. Furthermore, enzymatic activity has been observed not only with dopamine but also with histamine and other biogenic amines as substrates (Richardt et al., 2003).

line 184, line 227, “drosophila”: Drosophila should be capitalized and italicized.

lines 192/193, “… there is little information of the function and expression of the GB54634 gene in honey bees”: Is anything known about the function or the expression pattern of the homologous protein in the model organism Drosophila? Is there information about this on FlyBase or FlyAtlas?

line 224, “Aumer et al., 2019”: Reference number! (24)?

line 272, “Browning and Browning 2016”: Not in references list!

line 406, “HÄRtel”: Type “Härtel”.

Reviewer #2: The authors have re-analyzed one of their previous data sets to identify genes associated with body colour and ovary activation in African honey bees. I appreciate the approach and enjoyed reading the paper. I do have some concerns that need to be addressed prior to publication.

Major comments:

1) Do the authors have (or need) a permit to use the samples they have collected? I noticed the authors are in the US but the samples come from South Africa.

2) Data Accessibility. The authors need to upload their raw genomic data (fastq) to a repository. The results can't be replicated otherwise. The authors uploaded a vcf file to dryad but, as far as I can tell, the phenotypic data is not uploaded and with the vcf . This should be included in the raw data upload.

3) Imputation. I think the authors need to discuss and test the impacts of imputation on their data set. I have specific points on this below but the authors imply that up to 92% of the genotypes at a site were imputed. It would be worthwhile to convince the reader that imputation didn't impact your overall findings and/or the extent to which results relied on imputation.

Specific Comments

Line 101 - perhaps not the case, maybe reference previous studies by Oldroyd and Moritz groups?

Line 117 - previous work identifying QTLs for ovary activation found them on LG 11 (Linksvayer, Page, and others). Could you highlight why you're not finding an association here? Perhaps in your discussion? It's likely because of different populations being studied but this is still something that should be highlighted.

Line 117 - it would be unclear to the non-bee expert which subspecies is which color and/or has what ovariole count. Perhaps this can be elaborated on in a figure? Additionally, it would be worthwhile to see the distribution of associated SNPs within each subspecies (with and without imputation). Were these sites also found to have high Fst in previous, referenced, studies?

Line 117 - You don't elaborate much on the overlap among the traits. Do you have a quantitative relationship between body color index and ovary number? Does this relationship also associate with the associated GWAS sites.

Line 127 - your significance cut off is not explained in detail here nor in the methods. Is Bonferroni appropriate here and did you select the correct number of independant comparisons? You likely have fewer independent comparisons than you have used.

Line 165- I don't know if I would say that coloration is not a trait of interest in beekeepers. In the US, beekeepers have a long history of coloration preference in their bees (e.g. the 'three gold lines' that were preferred in Italian stocks, during early importation).

Line 168 - I'm not an ecologist, but is this Bergmann's rule? I thought that was for body size? It might be Gloger's Rule?

Line 174 - Is (24) the correct reference? Did the authors intended this one: doi.org/10.1093/molbev/msz100

Line 269 - It seems striking to allow a site to have 92% of the data and be imputed. I am curious to know the effect of imputation on this data set?

Line 271 - the authors don't explain FIT nor why they are trimming based on it.

Line 274 - 11,656 SNPs retained per individual seems like very low coverage for a genome with such a high recombination rate and very sparse for imputation given the high recombination rate. With so few SNPs per individual, how do the authors extract associated sites across all individuals (Table 1)?

Figure 4 has no description. Figure 2 and 3 descriptions seem to be swapped.

6. PLOS authors have the option to publish the peer review history of their article (what does this mean?). If published, this will include your full peer review and any attached files.

Reviewer #1: No

Reviewer #2: No

---

## [Author Response · Author response to Decision Letter 0]

3 Nov 2021

Reviewer #1: This is an interesting paper on a genome wide association study in Apis mellifera capensis, Apis mellifera scutellata and hybrid individuals looking for associations to the number of ovarioles and two color-related traits: tergite and scutellar plate color. The authors discovered two possible genetic variations associated to these traits, a deletion in the ebony gene and a premature stop on gene GB54634.The experiments seem to be performed accurately and are well presented. The subject is put into the context of similar studies and is discussed appropriately. In my opinion, the manuscript would make a useful addition to the literature. Only a few minor suggestions to further improve the manuscript came into my mind.

R: We would like to thank reviewer #1 for the thoughtful and much appreciated comments throughout the review process. 

Specific comments:

Minor issues:

line 102, “Bustamente et al. (2020)”: Enter the reference number as in the references list! (21)? See also line 249.R: We addressed this comment as suggested. 

line 108, “Eimanifar et al. (2018a)”: Reference number! (18)?

R: We addressed this comment as suggested. 

line 137 or elsewhere: When the Ebony protein is introduced, it should be mentioned that Ebony is a non-ribosomal peptide synthetase, which also has sequence similarities to microbial enzymes (Hovemann et al., 1998).

R: We have added the information, thank you for your suggestion. This is a very good reference.

lines 144/145, line 147, “Figure 2”: Check Figure numbers! Protein structures are given in Figure 3.

line 153, line 157, “Figure 3”: Check Figure numbers! Description matches Figure 2.

R: Thank you, we have fixed the figure labels accordingly.

lines 158/159, “Ruttner (1988)” and “Bustamente et al., (2020)”: Enter reference numbers as in the references list! (2) and (21), respectively?

R: Edited as suggested

lines 177-178, Variants in Ebony also contribute to diverse phenotypic variations including … visual ability: In fact, it was shown some time ago that the electroretinogram (ERG) of certain Drosophila ebony mutants lacked the on- and off-transients of the light response (Hotta and Benzer, 1969; Heisenberg, 1971).

R: Thank you for the suggestion, we have added it to the text – this is interesting and possibly relevant for the manuscript.

lines 178-180, “Ebony participates in dopaminergic neuron function, metabolizing dopamine into N-β-alanyl dopamine (NBAD)”: That is correct, but not the whole truth. Drosophila ebony mutants also have reduced histamine content in the head and are unable to convert histamine into its β-alanine conjugate, carcinine (Borycz et al., 2002). In vitro assays with heterologously expressed and purified Drosophila Ebony protein have shown later that the activation and binding of β-alanine occurs in a peptide-synthetase like manner. Furthermore, enzymatic activity has been observed not only with dopamine but also with histamine and other biogenic amines as substrates (Richardt et al., 2003).

R: Thank you for the information you provided! We would like to include it in the text; yet, to maintain the logical flow and make the information presented in the manuscript more direct, we decided to prioritize the influence of Ebony on dopamine, as this is a crucial part of the behavioral modifications cited by other authors. 

line 184, line 227, “drosophila”: Drosophila should be capitalized and italicized.

R: We addressed this comment as suggested.

lines 192/193, “… there is little information of the function and expression of the GB54634 gene in honey bees”: Is anything known about the function or the expression pattern of the homologous protein in the model organism Drosophila? Is there information about this on FlyBase or FlyAtlas?

R: Unfortunately, it is the case for all other model organisms (and non-model Insecta) we have researched. It is an uncharacterized gene with no further information that could provide us any clues on its function https://useast.ensembl.org/Drosophila_melanogaster/Gene/Summary?db=core;g=FBgn0034808;r=2R:22993669-22998884;t=FBtr0071988

http://flybase.org/reports/FBgn0034808.html

line 224, “Aumer et al., 2019”: Reference number! (24)?

R: We addressed this comment as suggested.

line 272, “Browning and Browning 2016”: Not in references list!

R: We addressed this comment as suggested.

line 406, “HÄRtel”: Type “Härtel”.

R: We addressed this comment as suggested.

Reviewer #2: The authors have re-analyzed one of their previous data sets to identify genes associated with body colour and ovary activation in African honey bees. I appreciate the approach and enjoyed reading the paper. I do have some concerns that need to be addressed prior to publication.

R: We would first like to thank reviewer #2 for the thoughtful comments. ! 

Major comments:

1) Do the authors have (or need) a permit to use the samples they have collected? I noticed the authors are in the US but the samples come from South Africa. 

R: A permit was not needed given we sampled managed colonies, with permission from the beekeeper owners. (See also the more detailed response to this question among the queries from Reviewer 1).

2) Data Accessibility. The authors need to upload their raw genomic data (fastq) to a repository. The results can't be replicated otherwise. The authors uploaded a vcf file to dryad but, as far as I can tell, the phenotypic data is not uploaded and with the vcf . This should be included in the raw data upload.

R: Thank you for noting this. We have attached the phenotype file with the manuscript. Unfortunately, for GBS studies, the file generated by the service provider (Cornell University Life Sciences Core Facility) at the time was a VCF, and not a series of fastqs representing each individual. This is the reason we uploaded the vcf. We sincerely hope this is sufficient, as we have also used the vcf (PLINK can convert VCFs to a .bed format for analysis) for our own analysis.

3) Imputation. I think the authors need to discuss and test the impacts of imputation on their data set. I have specific points on this below but the authors imply that up to 92% of the genotypes at a site were imputed. It would be worthwhile to convince the reader that imputation didn't impact your overall findings and/or the extent to which results relied on imputation.

R: Thank you for your concern with imputation. Yet, we believe the reviewer might have misinterpreted the materials and methods, as the value referenced of 92% is the threshold for missing data (missing calls or genotypes) for each SNP. This means each SNP that passed had at least 92% or more samples successfully genotyped, and this step is prior to imputation.We will address this further below.

Specific Comments

Line 101 - perhaps not the case, maybe reference previous studies by Oldroyd and Moritz groups?

We agree with the statement. We modified the text to read:

Despite genomic studies on various honey bee subspecies, the genetic basis of important phenotypes found in A.m. scutellata and A.m. capensis remain poorly understood, though progress has been made with the thelytoky trait [19-25]. 

Line 117 - previous work identifying QTLs for ovary activation found them on LG 11 (Linksvayer, Page, and others). Could you highlight why you're not finding an association here? Perhaps in your discussion? It's likely because of different populations being studied but this is still something that should be highlighted.

R: Thank you for your comment. We have added a comment on this study in the discussion.

“Previous work evaluating quantitative trait loci (QTLs) impacting the number of ovarioles in honey bees resulted in a significant QTL on LG11 [43]. Although our GWAs did not associate any markers on LG11 to ovariole number, this difference in findings could be due to population genetic differences as the LG11 QTL resulted from Africanized Honey Bees (AHB) collected in Arizona, USA, compared to European Honey Bee samples collected from US commercial colonies.”

Line 117 - it would be unclear to the non-bee expert which subspecies is which color and/or has what ovariole count. Perhaps this can be elaborated on in a figure? Additionally, it would be worthwhile to see the distribution of associated SNPs within each subspecies (with and without imputation). Were these sites also found to have high Fst in previous, referenced, studies?

R: Thank you for this comment – In the manuscript Introduction, we present information on the coloration and form of reproduction for both subspecies. Figures 3e and 4b/c demonstrate the variation in coloration, and Figure 4a, the variation in ovariole number by subspecies; and we have included the respective genotypes per individual sample in the supplemental file 1, as well as a S2 file with the allelic distribution per subspecies. As for the distribution of associated SNPs , polymorphisms within coding regions were discovered and analyzed based on data from the unfiltered (not imputed) GBS generated .vcf file (Material and Methods). . We did not conduct a population analysis in this study, as this was previously done and published in Eimanifar, A., Brooks, S., Bustamante, T. et al. Population genomics and morphometric assignment of western honey bees (Apis mellifera L.) in the Republic of South Africa. BMC Genomics 19, 615 (2018). https://doi.org/10.1186/s12864-018-4998-x

A full description of the Fst analysis can be found in in the referenced manuscript.

Line 117 - You don't elaborate much on the overlap among the traits. Do you have a quantitative relationship between body color index and ovary number? Does this relationship also associate with the associated GWAS sites.

R: We have performed a correlation analysis between the color traits and the ovary number, and both demonstrate a significant negative correlation (lighter individuals have fewer ovarioles; P<.0001); Yet as for the associated variants, the correlation was only significant between color and the NC_007070.3:g.14101325G>del;p.R69Efs*85 variant ( P<.0001); and between ovary number and the NC_007076.3:g.4492792A>G;p.Tyr128* variant (P=0.0003). We can add this analysis as a supplemental file if the reviewer deems necessary. 

Line 127 - your significance cut off is not explained in detail here nor in the methods. Is Bonferroni appropriate here and did you select the correct number of independant comparisons? You likely have fewer independent comparisons than you have used.

R: Thank you pointing this out; we had a typo in the text. We had entered the log value for Bonferroni in the text. Bonferroni is a strict multiple testing cut frequently utilized in conservative studies with a large number of variables as in genome wide association analysis. We utilized the number of traits (3) and markers (11,656) to correct for a value of alpha of 0.05/34968 = 1.429 x 10-6 . 

Line 165- I don't know if I would say that coloration is not a trait of interest in beekeepers. In the US, beekeepers have a long history of coloration preference in their bees (e.g. the 'three gold lines' that were preferred in Italian stocks, during early importation).

We agree. We deleted this text from the manuscript. 

Line 168 - I'm not an ecologist, but is this Bergmann's rule? I thought that was for body size? It might be Gloger's Rule?

R: Thank you for pointing it out, we have corrected the text accordingly.

Line 174 - Is (24) the correct reference? Did the authors intended this one: doi.org/10.1093/molbev/msz100

R: Thank you for pointing it out, we have added this reference. The original reference also mentions the information in the paragraph, yet we appreciate the different perspective of the suggested paper.

Line 269 - It seems striking to allow a site to have 92% of the data and be imputed. I am curious to know the effect of imputation on this data set?

R: Thank you for your concern with imputation. Yet, we believe the reviewer might have misinterpreted the materials and methods, as the value referenced of 92% is the threshold for missing data (missing calls or genotypes) for each SNP. The number of imputed SNPs was 1600 SNPs; and these went through a second round of QC with a MAF threshold of 5%, leaving about 11,656 SNPs that were used for the GWAs. 

Line 271 - the authors don't explain FIT nor why they are trimming based on it.

R: According to the findings of three published studies performed by Eimanifar et al. (References No. 18, 47, and 48), the null hypothesis of panmixia could not be rejected because the random distribution of genotypes was determined within multiple localities of the A.m. capensis population where no discernible geographic zone was recognized between two subspecies. Although the SNP data in reference no. 18 support the existence of a distinct population within A.m. scutellate, the magnitude of the panmictic gene pool across the entire geographical sampling area should not be underestimated. As a result, A.m. capensis had a large population size, with admixture evidence in the sampling areas. Therefore, we applied the panmixia index with the lowest effect to retain the most appropriate sequencings per individual for downstream analyses.

Line 274 - 11,656 SNPs retained per individual seems like very low coverage for a genome with such a high recombination rate and very sparse for imputation given the high recombination rate. With so few SNPs per individual, how do the authors extract associated sites across all individuals (Table 1)?

R: Indeed, the coverage is lower than ideal, but what was feasible given the GBS approach as originally budgeted for a smaller scale marker set suitable for the population work conducted in Eimanifar et al 2018. 11,656 SNPs is after a second round of QC and trimming. The GBS approach initially aimed to produce about 70k markers across the genome of these two subspecies. However, coverage using GBS is not at all uniform, and after applying stringent quality filters for the GWAs (32998 lost during filtering for MAF and 50 lost due to missingness) only 11,656 SNP remained). This coverage is about 1 SNP per 19,818bp. Genome-wide LD averaged 0.035 (r2) using this panel, and is comparable to that measured in other studies. The honey bee genome (~231 million bps) is considerable smaller than an average mammal species

Figure 4 has no description. Figure 2 and 3 descriptions seem to be swapped.

R: Thank you, we have fixed the figure labels. The description of figure 4 is located later in the Material and Methods, and the reviewer might not have seen it. It is: 

Fig 4. Distribution of morphometric phenotypic traits per subspecies, representing a. Ovariole Number quantile, b. Tergite Color quantile and c. Scutellar Plate color ranking. Red represents increased number of ovarioles (a) or lighter phenotypes (b and c), while blue represents lower number of ovarioles (a) and darker phenotypes (b and c). The visual distribution seems to correlate to the subspecies or hybrid geographical distribution.

---

## [Editor Report · Decision Letter 1]

18 Nov 2021

Attack of the Dark Clones: The genetics of reproductive and color traits of South African honey bees (Apis mellifera spp.)

PONE-D-21-27869R1

Dear Dr. Patterson Rosa,

We’re pleased to inform you that your manuscript has been judged scientifically suitable for publication and will be formally accepted for publication once it meets all outstanding technical requirements.

Kind regards,

Olav Rueppell

Academic Editor

PLOS ONE
---

## [Editor Report · Acceptance letter]

1 Dec 2021

PONE-D-21-27869R1 

Attack of the Dark Clones
The genetics of reproductive and color traits of South African honey bees (*Apis mellifera* spp.) 

Dear Dr. Patterson Rosa:

I'm pleased to inform you that your manuscript has been deemed suitable for publication in PLOS ONE. Congratulations! Your manuscript is now with our production department. 

Kind regards, 

on behalf of

Dr. Olav Rueppell 

Academic Editor

PLOS ONE